# Heat Stress Mitigation: Impact of Increased Cooling Sessions on Milk Yield and Welfare of Dairy Buffaloes in a Semiarid Summer

**DOI:** 10.3390/ani13213315

**Published:** 2023-10-25

**Authors:** Syed Israr Hussain, Nisar Ahmad, Saeed Ahmed, Maqsood Akhter, Muhammad Qamer Shahid

**Affiliations:** 1Department of Livestock Management, University of Veterinary and Animal Sciences, Lahore 54000, Pakistan; israrvet@gmail.com (S.I.H.); nisarahmad@uvas.edu.pk (N.A.); 2Department of Animal Nutrition, University of Veterinary and Animal Sciences, Lahore 54000, Pakistan; saeed.ahmed@uvas.edu.pk; 3Livestock Production Research Institute, Okara 56301, Pakistan; drmaqsood66@gmail.com

**Keywords:** dairy buffaloes, heat stress, sprinkler cooling, milk production, physiological measures

## Abstract

**Simple Summary:**

In this study, we assessed the impact of increased cooling sessions on milk production and physiological measures, including respiration rate and body temperature, in dairy buffaloes during a semiarid summer in Pakistan. Eighteen Nili Ravi buffaloes were assigned to three cooling strategies: CTL (handheld hosepipe cooling twice daily for 5 min), 3CS (sprinkler cooling three times daily), and 5CS (sprinkler cooling five times daily). The 5CS group produced 1.6 and 3.2 kg more milk per day than 3CS and CTL, respectively, with more milk fat. The buffaloes in the 5CS group also had lower core body temperatures and respiration rates than the CTL and 3CS groups. Furthermore, buffaloes in the 5CS group spent significantly more time eating and showed decreased blood cortisol levels. Our study demonstrated that increased cooling sessions had a beneficial effect on milk yield and composition, particularly by increasing milk fat content. Additionally, more cooling sessions positively influenced the welfare of dairy buffaloes by lowering respiration rate, body temperature, and cortisol levels. These findings underscore the critical importance of optimizing cooling practices in dairy buffalo management.

**Abstract:**

The current study aimed to evaluate the impact of increasing cooling sessions from three to five times a day on milk yield and the welfare of dairy buffaloes during a semiarid summer in Pakistan. Eighteen Nili Ravi buffaloes were randomly assigned to three cooling strategies: (1) CTL, where buffaloes were cooled with a handheld hosepipe twice daily for 5 min each; (2) 3CS, where buffaloes were cooled using sprinklers three times daily; and (3) 5CS, where buffaloes were cooled using sprinklers five times daily. Each sprinkler cooling session lasted 1 h, with a 6 min cycle of 3 min of water on and 3 min off. Results showed that the 5CS group produced 1.6 and 3.2 kg more milk per day compared to the 3CS and CTL groups, respectively (*p* < 0.001). Both the 5CS and 3CS groups had consistently lower core body temperatures and respiration rates than the CTL group. Buffaloes in the 5CS group spent significantly more time eating (*p* < 0.001). Additionally, the 5CS group exhibited lower cortisol and blood urea nitrogen levels (*p* = 0.001) and higher glucose levels than the CTL group (*p* = 0.006). In conclusion, increasing cooling sessions to five times daily improved milk yield and welfare compared to the traditional cooling strategy (CTL) in dairy buffaloes during semiarid summers, highlighting the benefits of optimized cooling practices.

## 1. Introduction

Water buffaloes are resilient and versatile animals that play a vital role in agricultural economies worldwide, providing milk, meat, skin, and farm power [1]. They are the primary dairy animals in countries such as India and Pakistan [2], and they also hold significant importance in specialized niches, such as the production of Mozzarella cheese and other innovative products in European countries [3]. However, as temperatures rise due to climate change, these animals face substantial challenges to their well-being and productivity due to heat stress [4]. While buffaloes have adapted well to hot and humid environments, their method of heat dissipation differs from that of cattle. They have fewer sweat glands, which results in less efficient evaporative cooling [5]. Additionally, during heat stress, buffaloes exhibit a relatively lower respiratory rate compared to cattle [4] and demonstrate less effective respiratory evaporative cooling [6]. However, their low hair density enables them to dissipate heat through convection and radiation rather than relying on evaporation [7]. This is why buffaloes rely on wallowing in water as a natural method to alleviate heat load [6], facilitated by increased blood vessels in the skin that help irradiate heat [8]. The increasing intensification of buffalo farming systems [9] and the decreasing availability of potholes and ponds in rural areas have led to restricted or even nonexistent opportunities for water buffaloes to engage in wallowing [4,10]. Consequently, this has heightened the negative impacts on these animals due to heat stress, making them increasingly vulnerable in the coming years. Increased environmental temperatures affect the milk production, growth rate, and reproductive performance of water buffaloes [5,11].

In countries such as Pakistan, farmers employ handheld hoses to spray water on buffaloes to cool them down during the summers. This practice is conducted twice a day, with brief sessions in the morning and afternoon, each lasting just a few minutes [10]. While this method is commonly used, it consumes a significant amount of groundwater and offers limited relief from heat stress compared to sprinkler cooling systems [12]. Sprinklers are potentially more effective at cooling buffaloes compared to the traditional method of using hoses to spray water. This is because the duration for which the water spraying is active, and the resulting water drippings, help remove heat from the animal’s body [13] through convection, thereby increasing heat dissipation. Applying sprinklers in sessions can provide cooling for longer durations. Recent research has highlighted the potential of sprinkler cooling as a valuable alternative [12]. The study revealed that increasing the frequency of sprinkler sessions from two to three resulted in a lower respiration rate and an increase in milk and milk fat yield, underscoring the positive impact of enhanced cooling sessions [12].

The objective of the current study was to assess the impact of increasing the frequency of sprinkler cooling sessions from three to five times a day on milk yield and the welfare of dairy buffaloes in the semiarid summers of Pakistan.

## 2. Materials and Methods

The experiment was conducted at the Livestock Experimental Station in Haroonabad, Punjab, Pakistan (N 29 38.674349′, E 73 9.016675′), during the summer of 2022 (June and July). 

### 2.1. Study Animals, Housing, and Management 

A total of eighteen (18) Nili Ravi buffaloes with an average daily milk yield of 5.0 ± 0.4 kg, days in milk of 131 ± 64, and parity of 3.6 ± 2.1 (mean ± SD) were selected for the study. The enrolled buffaloes were healthy, showing no clinical signs of mastitis, lameness, or any other health issues. The buffaloes were kept in a naturally ventilated shed with a concrete floor and a precast concrete roof, approximately 3.7 m in height. The shed measured 21 m in length (east-west) and 10 m in width, with both the south and north sides having large openings for ventilation, and an adjacent loafing area with a brick floor. The buffaloes were tethered at the manger using neck chains during the treatment application period (7:00 to 16:00 h) while they had free access to the adjacent loafing area outside during the night. A sprinkler line, equipped with nozzles positioned to spray water at each tied animal, was installed. Each nozzle was fitted with a valve to control the flow rate. The daily ration consisted of fresh sorghum fodder and concentrate at 50 kg and 3 kg per buffalo, respectively. Each buffalo was provided with a water tub placed near it in the shed during the daytime, and a water trough, in the loafing area, was available for free access during the nighttime. The water tubs and troughs were kept filled for ad libitum water intake. Manual milking was conducted twice daily in a separate shed at 05:00 and 17:00 h. 

### 2.2. Experimental Design and Study Groups

Eighteen Nili Ravi buffaloes were randomly divided into three groups of 6 buffaloes each. The buffaloes were then assigned to three cooling session strategies. The cooling strategies consisted of (1) CTL, where buffaloes were cooled by applying water using a handheld hosepipe twice daily, for 5 min each time, with a water flow rate of approximately 40 L/min; (2) 3CS, where buffaloes were cooled using sprinklers three times daily; and (3) 5CS, where buffaloes were cooled using sprinklers five times daily. The cooling in the CTL group was applied at 07:00 and 15:00 h. The sprinkler sessions for the 3CS group were scheduled at 07:00, 11:00, and 15:00, while the 5CS group had two additional sessions at 09:00 and 13:00 h. In the case of 3CS and 5CS, each sprinkler cooling session lasted for 1 h with a 6 min cycle consisting of 3 min of water on and 3 min off. The sprinkler flow rate was set at 2 L/min. The sprinkler flow rate and cycle were selected based on the findings of an initial study on buffaloes [14]. The 1 h sprinkler duration was a strategy aimed at using intermittent sessions rather than continuous sessions to reduce water consumption. A polyvinyl water pipe, equipped with sprinkler nozzles, was installed at a height of approximately 2.1 m above the floor, positioned along the manger, and situated roughly 50 cm away from it. In the 3CS and 5CS groups, the tethering of each buffalo during the daytime was positioned in alignment with a sprinkler nozzle. These sprinkler nozzles had a 180-degree angle, directing water towards the back of the buffalo, and providing cooling to the withers, back, and main trunk. The spread of the sprinklers extended 1.8 m along the manger and 1.8 m away from the manger, covering an area of approximately 3.24 square meters. The showering cycles were controlled through an automated valve installed in the showering line for each group, which was operated by a programmable logic control panel (Wecon Technology; Model: Levi 2070D; Version: VI.2.4.1.7.2.0; Fuzhou, China). The treatment applications were implemented over a period of six weeks. 

### 2.3. Climate Measures

Environmental measures, such as air temperature (T, °C), relative humidity (RH, %), temperature–humidity index (THI), black globe temperature (BGT, °C), heat load index, and wind speed, were collected using a portable weather station (model: Kestrel 5400 Cattle Heat Stress Tracker: 0854AGLVCHVG). The weather station was placed in the outside loafing area and configured to record climate measurements at a 20 min interval each day throughout the study period.

### 2.4. Production, Physiological, and Behavioral Measures

Milk yield was recorded daily both for morning and evening milking sessions. 

Weekly milk composition analysis was conducted using a portable milk analyzer (Model: Lactoscan S, Milktronic Ltd., Nova Zagora, Bulgaria). Approximately 20 mL of fresh milk samples of each buffalo from both morning and evening milking sessions were collected from the middle of the milking bucket after thorough mixing. These samples were promptly analyzed within 5 min of milking. The milk was analyzed for protein, fat, and lactose content.

A subset of three buffaloes from each group was randomly selected for core body temperature (CBT) measurement. The CBT of the buffaloes was recorded every 20 min for two consecutive days during each week using thermochron data loggers (iButton: model DS1921H-F5, iButtonLink, Llc., Whitewater, WI, USA). The data loggers were placed intravaginally using an inert controlled internal drug release insert (Zoetis, Auckland, New Zealand). The rectal temperature (RT) and respiration rate (RR) of all enrolled buffaloes were recorded at five different time points: approximately 07:00, 08:00, 11:30, 14:30, and 15:30 h. The RT was recorded using a digital thermometer while the RR was measured by visually observing flank movements and counting them to determine the number of breaths per minute. The measurements of the RR and RT were consistently taken by the same individual.

Behavioral measures, such as daily lying time, eating time, and standing time, were recorded using the Nedap CowControl^TM^ system (NEDAP, Groenlo, The Netherlands). Eating time was tracked via neck collars while standing and lying times were monitored using leg data loggers. These measures were continuously monitored for 24 h throughout the study.

### 2.5. Blood Metabolites

Blood samples were collected once a week from the jugular vein, specifically during the morning hours at approximately 11:00 h. The samples were collected into sterile anticoagulant vacutainer tubes, promptly maintained within a cold chain, and transported to the laboratory. In the lab, the samples underwent centrifugation, resulting in the separation of serum, which was then stored at −20 °C until analysis. The serum was subjected to analysis for glucose (utilizing the Glucose GOD FS kit from DiaSys Diagnostic Systems GmbH, Holzheim, Germany), blood urea nitrogen (employing the Randox Urea Kinetic kit from Randox Laboratories Ltd., Crumlin, UK), and cortisol (using the Cortisol Elisa kit from Calbiotech, California, USA), with the aid of a spectrophotometer (Epoch2, BioTek, Winooski, VT, USA).

### 2.6. Statistical Analysis

Statistical analyses were performed using SAS (SAS for Academics: SAS Institute Inc., Cary, NC, USA). The data were assessed for normality according to the Shapiro–Wilk test. The RT, RR, milk yield, and behavioral data were averaged to weekly means and then subjected to a repeated-measures ANOVA using the Mixed Procedure of SAS. The Least square means were separated using the PDIFF option with Tukey’s adjusted *p*-values. Differences were considered significant at *p* ≤ 0.05 and tendencies at *p* < 0.10.

## 3. Results

### 3.1. Environmental Measures

The average air temperature (T), temperature–humidity index (THI), heat load index (HLI), black globe temperature (BGT, °C), relative humidity (RH, %), and wind speed (WS, m/s) are summarized in Table 1. During the treatment period, the mean T and RH were 40 °C and 37%, respectively, indicating a hot environment. The THI, during the treatment period, was 86.5 ± 4.2 (mean ± SD).

### 3.2. Production, Physiological, and Behavioral Measures

The results demonstrated a significant influence of cooling session strategies on both milk yield and milk components. Buffaloes in the 5CS group exhibited a higher milk production, producing an average of 1.6 and 3.2 kg more milk per day compared to the 3CS and CTL groups, respectively (8.5 vs. 6.9 and 5.3 kg/day, respectively; SE = 0.2; *p* < 0.001; Table 2). Similarly, buffaloes in the 5CS group had higher fat content, with 2 and 2.9% higher fat compared to the 3CS and CTL groups (8.1 vs. 6.1 and 5.1%, respectively; SE = 0.1; *p* < 0.001; Table 2). Interestingly, the protein and lactose percentages were lower in the 5CS group compared to the 3CS and CTL groups. All the recorded milk components, measured in grams, were higher in the 5CS group compared to the other groups. 

The hourly core body temperature (CBT) values over a 24 h period are presented in Figure 1. The cooling session strategies had a significant impact on the CBT. Specifically, the CBT values were consistently lower in both the 5CS and 3CS groups compared to the CTL group during the treatment application period from 07:00 to 15:00 h. Likewise, the 5CS group exhibited the lowest respiration rate (RR) and rectal temperature (RT), followed by the 3CS group and the CTL group, as depicted in Figure 2 and Figure 3, respectively. The buffaloes in the 5CS group exhibited an average of 0.5 and 0.3 °C lower RT during the afternoon hours compared to the CTL and 3CS groups, respectively. Similarly, the buffaloes in the 5CS group showed an average of 1.9 and 5.3 breaths/min lower respiration rate during the afternoon hours compared to the CTL and 3CS groups, respectively. The average respiration rate RR of buffaloes in the 5CS group was 18.8, while it was 23.3 in the CTL group.

The results revealed that cooling session strategies had a significant impact on eating behavior. Buffaloes in the 5CS group exhibited a significantly longer duration of eating, spending an average of 83 and 111 min per day more than those in the 3CS and CTL groups, respectively (224 vs. 113 and 142 min/day, respectively; SE = 7.4; *p* < 0.001; Table 3). However, the cooling strategies did not have a noticeable effect on total rumination time, total lying time, and total step counts (Table 3). On average, buffaloes spent 7.8 ± 0.2 h lying, 8.8 ± 0.42 h in rumination, and recorded 5.93 ± 0.37 (×1000) number of steps per day. Notably, the total inactive standing time was numerically lower in buffaloes within the 5CS group when compared to the CTL group (9.8 vs. 11.3 h/day; SE = 0.52; *p* < 0.105; Table 3).

### 3.3. Blood Metabolites

The current findings demonstrated a significant impact of cooling session strategies on all the measured blood metabolites. Buffaloes in the 5CS group exhibited lower cortisol levels compared to the 2CS and CTL groups, with 0.9 and 1.7 µg/dL less cortisol, respectively (3.8 vs. 4.7 and 5.5 µg/dL, respectively; SE = 0.37; *p* = 0.002; Table 4). Blood glucose levels were higher in the 5CS group compared to the CTL group (89.4 vs. 71.8 mg/dL; SE = 4.0; *p =* 0.006; Table 4). Both the 5CS and 3CS groups showed similar blood glucose levels (Table 4). Furthermore, the highest blood urea nitrogen levels were observed in the CTL group, followed by the 3CS and 5CS groups. 

## 4. Discussion

### 4.1. Climate Measures

The summer period in the study area was characterized by high temperatures, causing heat stress for the buffaloes. The THI is generally used as an indicator of heat stress in dairy animals [15]. Climate measures in the current study demonstrated that the buffaloes experienced moderate to severe heat stress throughout the study period, with the average 24 h THI exceeding the threshold for dairy buffaloes [16]. This high level of heat stress provided an opportunity to effectively evaluate the impact of cooling interventions on buffalo performance.

### 4.2. Milk Yield and Composition

The findings of this study revealed significant effects of the cooling strategies on milk and milk component yield in buffaloes. The 5CS buffaloes exhibited higher milk yield compared to those in the 3CS and CTL groups, indicating the positive influence of more frequent and prolonged cooling sessions on milk production. The improved thermal comfort and reduced heat stress experienced by the buffaloes in the 5CS group likely contributed to their increased productivity. Additionally, the extended feeding time in the 5CS group may have played a role in the higher milk yield by allowing the buffaloes to consume more nutrients for increased milk production. A positive association between feed intake and milk yield is well established [17,18,19]. The lower milk yield in the CTL and 3CS groups could be attributed to lower dry matter intake and lower feed digestibility [20]. Several studies on cattle and buffaloes have demonstrated that increasing the frequency and duration of cooling sessions can improve milk yield under heat stress conditions [4,12,17]. These findings align with the results of this study, further emphasizing the positive impact of cooling interventions on milk production.

As for milk quality, the 5CS group showed a higher fat content in the milk compared to the 3CS and CTL groups, suggesting that the more cooling sessions in this group contributed to enhanced fat synthesis and accumulation in the mammary glands. It is widely acknowledged that an increase in milk yield is often accompanied by a decrease in milk fat percentage, attributed in part to the dilution effect [21,22]. This phenomenon holds true under normal conditions. However, under heat stress conditions, both milk yield and composition are negatively impacted [23]. In our study, the buffaloes with less cooling (CTL and 3CS groups) had lower milk and milk component yield compared to the 5CS group. The lower fat content in these groups may be attributed to reduced short- and medium-chain fatty acids synthesis in the udder, as heat stress tends to decrease the production of these fatty acids [24,25]. The buffaloes with more cooling (5CS) exhibited a notable increase in milk yield and an increase in milk fat content in response to the cooling environment. This finding was in line with the previous study on buffaloes under similar climate conditions, where they found that more cooling improved milk yield and milk fat content [12]. However, it is important to note that the 5CS group had lower protein and lactose percentages compared to the other groups. This contradicts previous findings that reported lower milk protein and lactose content in heat-stressed cows [26]. The underlying factors influencing this discrepancy require further investigation. 

Overall, the findings highlight the importance of implementing effective cooling strategies to optimize milk yield in buffaloes under heat stress conditions. 

### 4.3. Physiological Measures

The lower CBT observed in the 5CS group during the treatment application period demonstrated the efficacy of this cooling strategy. In contrast, the CBT of the CTL and 3CS groups remained significantly higher than the 5CS group, indicating inadequate cooling and inefficient heat dissipation. These results align with previous studies [27,28], highlighting the importance of effective cooling strategies in overcoming the negative effects of heat stress on dairy animals.

The respiration rate, another indicator of heat stress, was recorded at different time points during the treatment application period in a day. The increased respiration rate observed at the start of the day in the CTL group of buffaloes could be attributed to the accumulated heat load from the previous day, which resulted from inefficient cooling. In contrast, the 5CS group showed no visible increase in respiration rate, indicating successful cooling and maintenance of normal respiratory function. These findings are consistent with the literature, as higher respiration rates are commonly observed in buffaloes exposed to heat stress [20,21,22,23,24,25,26,27,28,29]. The respiration rate observed in our study aligns with the reported range documented in the literature [12,30,31]. It is worth noting that buffaloes in the 5CS group exhibited an average respiration rate that was 1.9 and 5.3 breaths/min lower compared to the CTL and 3CS groups, respectively. This relatively small decrease in the RR suggests that buffaloes may not heavily rely on respiratory evaporative cooling, unlike cattle [6]. 

### 4.4. Behavioral Measures

Buffaloes in the 5CS group spent more time eating compared to those in the 3CS and CTL groups. The increased time spent eating in the 5CS group could be attributed to their improved thermal comfort and reduced heat load, enabling them to allocate more time and energy to feeding activities. This suggests that the more frequent and prolonged cooling sessions in the 5CS group positively influenced their feeding behavior. The findings regarding eating behavior are consistent with the notion that buffaloes tend to prioritize feeding when provided with a more favorable thermal environment. On the other hand, the higher inactive standing time observed in the CTL and 3CS groups may indicate their need for heat dissipation, as standing without eating is a common behavior observed in animals trying to dissipate heat [28]. Interestingly, no significant differences were observed among the treatment groups in other behavioral measures such as rumination time, lying time, step counts, and standing bouts. The lack of significant differences in step counts among the treatment groups can be attributed to certain factors. Firstly, it should be noted that the buffaloes were tied during the day, limiting their movement and ability to take long steps. Additionally, the similar step counts observed among the groups could be attributed to the comparable distances traveled during routine activities such as moving to the milking area and engaging in socialization during the relatively cooler hours of the day. Moreover, similar resting times among the groups could also contribute to the similarity in step counts. The tying of the buffaloes is a common practice, especially in small-scale dairy farming, which restricts their movement for management purposes.

Exploring additional behavioral parameters and considering factors such as social interactions and environmental enrichment could provide further insights into the behavioral effects of cooling interventions. Such knowledge can contribute to the development of more effective cooling strategies and the promotion of better buffalo management practices, ultimately enhancing their well-being and productivity in heat stress conditions. 

### 4.5. Blood Metabolites

The analysis of blood metabolites provided valuable insights into the physiological responses of buffaloes to heat stress and the effectiveness of cooling interventions. The results revealed that the 5CS group had lower cortisol levels, indicating reduced stress and better adaptation to heat stress conditions. This is consistent with the previous studies that have reported higher cortisol levels in ruminants under heat stress [12,29]. This suggests that the more frequent and prolonged cooling sessions in the 5CS group were effective in mitigating the negative effects of heat stress. On the other hand, the CTL group showed higher levels of blood urea nitrogen (BUN), which suggests a potential metabolic imbalance associated with heat stress. This finding is consistent with previous studies that have reported higher BUN levels in animals with less cooling [12]. In dairy cows, it has been observed that they preferentially use more glucose during heat stress, and the liver increases gluconeogenesis from amino acids to provide glucose [32]. The increased BUN levels in cows under heat stress could be a result of excessive deamination to provide amino acids for gluconeogenesis [33]. 

Interestingly, the 5CS group exhibited higher blood glucose levels compared to the CTL group. This could be indicative of improved energy metabolism and nutrient utilization in the buffaloes subjected to the 5CS cooling strategy. This finding aligns with previous studies that have reported decreased blood glucose levels under heat stress conditions [12,16,34]. The higher feeding time in the 5CS group could also be a contributing factor to the observed higher blood glucose levels.

### 4.6. Study Limitations

The study has some limitations, including a relatively small sample size and a single geographic location, potentially limiting the generalizability of the findings. Additionally, the study primarily focused on short-term effects during the summer season, without considering long-term implications or economic feasibility. Implementing enhanced cooling strategies on dairy farms may require infrastructure investment and ongoing operational costs, which need careful consideration. The integration of solar energy could potentially help mitigate these operational expenses in the long run. 

## 5. Conclusions

In conclusion, this study demonstrates the positive impact of cooling strategies on the welfare, productivity, and physiological responses of buffaloes under heat stress conditions. The 5CS cooling strategy, involving more frequent and prolonged cooling sessions, effectively mitigated heat stress, improved physiological parameters, enhanced feeding behavior, and increased milk yield. These findings emphasize the importance of implementing suitable cooling measures to optimize buffalo welfare and productivity in similar climatic conditions. 

## Figures and Tables

**Figure 1 animals-13-03315-f001:**
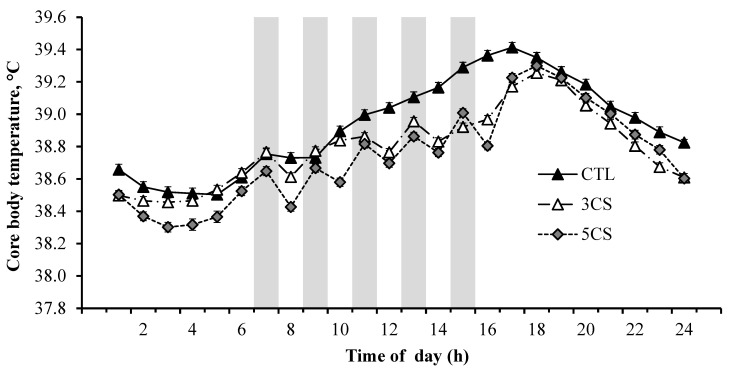
Average hourly core body temperature in Nili Ravi buffaloes at various time points throughout the day under three different cooling session treatments (n = 3 buffaloes per treatment group, 12 d of recording/animal, 24 h/day). The cooling strategies were CTL, cooling buffaloes twice daily at 07:00 and 16:00 h for 5 min each applying water with a handheld hosepipe; 3CS, three showering sessions of one hour each at 07:00, 11:00, and 15:00 h; 5CS, five showering sessions of one hour each at 07:00, 09:00, 11:00, 13:00, and 15:00 h. The showering was applied in cycles with 3 min water on and 3 min off using a flow rate of 2 L/min. The shaded columns represent the periods of cooling sessions. Error bars represent standard error (SE). Statistical analysis was performed individually for each time point, and rather than presenting individual *p*-values, the standard error (SE) was utilized to depict the variation within each group. The overlap of the error bars indicates a lack of significant difference between the groups.

**Figure 2 animals-13-03315-f002:**
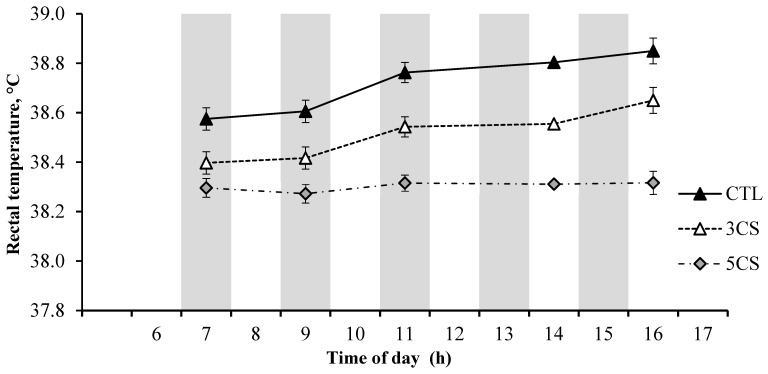
Average rectal temperature of Nili Ravi buffaloes at various time points throughout the day under three different cooling session treatments (n = 6 buffaloes per treatment group, 16 d of recording/animal, 5 times/day). The cooling strategies were CTL, cooling buffaloes twice daily at 07:00 and 15:00 h for 5 min each applying water with a handheld hosepipe; 3CS, three showering sessions of one hour each at 07:00, 11:00, and 15:00 h; 5CS, five showering sessions of one hour each at 07:00, 09:00, 11:00, 13:00, and 15:00 h. The showering was applied in cycles with 3 min water on and 3 min off using a flow rate of 2 L/min. The shaded columns represent the periods of cooling sessions. Error bars represent standard error (SE). Statistical analysis was performed individually for each time point, and rather than presenting individual *p*-values, the standard error (SE) was utilized to depict the variation within each group. The overlap of the error bars indicates a lack of significant difference between the groups.

**Figure 3 animals-13-03315-f003:**
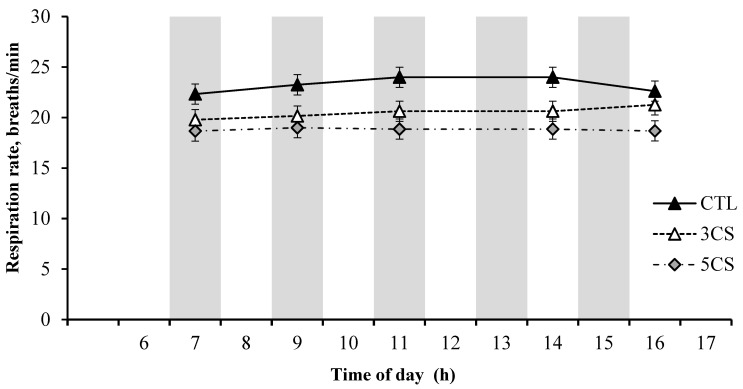
Mean respiration rate (breaths/min) of Nili Ravi buffaloes at various time points throughout the day under three different cooling session treatments (n = 6 animals per treatment group, 16 d of recording/animal, 5 times/day). The cooling strategies were CTL, cooling buffaloes twice daily at 07:00 and 15:00 h for 5 min each applying water with a handheld hosepipe; 3CS, three showering sessions of one hour each at 07:00, 11:00, and 15:00 h; 5CS, five showering sessions of one hour each at 07:00, 09:00, 11:00, 13:00, and 15:00 h. The showering was applied in cycles with 3 min water on and 3 min off using a flow rate of 2 L/min. The shaded columns represent the periods of cooling sessions. Error bars represent standard error (SE). Statistical analysis was performed individually for each time point, and rather than presenting individual *p*-values, the standard error (SE) was utilized to depict the variation within each group. The overlap of the error bars indicates a lack of significant difference between the groups.

**Table 1 animals-13-03315-t001:** Summary of daily environmental measures during June and July 2022.

Items	24 h			Daytime Period (07:00 to 17:00 h)		
	Mean ± SD	Minimum	Maximum	Mean ± SD	Minimum	Maximum
Temperature (T, °C)	36.0 ± 6.7	22.4	50.1	40.0 ± 6.6	22.4	50.1
Temperature–humidity index (THI)	83.0 ± 5.0	71.5	92.7	86.5 ± 4.2	71.9	92.7
Heat load index (HLI)	92.5 ± 11.7	68.2	117.9	102.6 ± 8.7	72.1	117.9
Black globe temperature (BGT, °C)	38.7 ± 9.3	22.3	58.3	46.1 ± 8.3	22.5	58.3
Relative humidity (RH, %)	41.1 ± 22	12	100	36.5 ± 25	12.2	100
Wind speed (WS, m/s)	0.14 ± 0.34	0	2.5	0.18 ± 0.38	0	2.5

**Table 2 animals-13-03315-t002:** Milk yield and composition of Nili Ravi buffaloes under varying cooling session treatments during a semi-arid summer.

Items	Cooling Sessions ^1^, n/d				
	CTL	3CS	5CS	SEM ^2^	*p* Value
Milk yield, kg/d	5.3 ^a^	6.9 ^b^	8.5 ^c^	0.2	<0.001
Milk components, %					
Fat	5.1 ^a^	6.1 ^b^	8.1 ^c^	0.1	<0.001
Protein	3.4 ^a^	3.3 ^b^	2.9 ^c^	0.04	<0.001
Lactose	4.9 ^a^	4.6 ^b^	4.2 ^c^	0.1	<0.001
Milk components, g/d					
Fat	273 ^a^	421 ^b^	678 ^c^	12	<0.001
Protein	183 ^a^	232 ^b^	243 ^c^	6	<0.001
Lactose	258 ^a^	314 ^b^	357 ^c^	10	<0.001

^a–c^ Values with different superscripts in a row are significantly different (*p* < 0.05). The values are presented as Least Square Means. ^1^ Cooling sessions treatment groups; CTL, cooling buffaloes twice daily at 07:00 and 16:00 h for 5 min each applying water with a handheld hosepipe; 3CS, three showering sessions of one hour each at 07:00, 11:00, and 15:00 h; 5CS, five showering sessions of one hour each at 07:00, 09:00, 11:00, 13:00 and 15:00 h. The showering was applied in cycles with 3 min water on and 3 min off using a flow rate of 2 L/min. ^2^ SEM = standard error of means.

**Table 3 animals-13-03315-t003:** Behavioral measures of Nili Ravi buffaloes under different cooling session treatments in a semi-arid summer.

Items	Cooling Session Treatments ^1^, n/d				
	CTL	3CS	5CS	SEM ^2^	*p* Value
Total feeding time, min/d	113 ^a^	142 ^b^	224 ^c^	7.4	<0.001
Total rumination time, h/d	8.1	9.3	8.9	0.42	0.152
Total lying time, h/d	7.9	7.9	7.6	0.2	0.701
Inactive standing, h/d	11.3	9.8	9.8	0.52	0.105
Standing bouts, number/d	8.7	8.2	8.2	0.29	0.352
Step count, ×1000 number/d	6.1	5.7	6.0	0.37	0.531

^a–c^ Values with different superscripts in a row are significantly different (*p* < 0.05). The values are presented as Least Square Means. ^1^ Cooling sessions treatment groups; CTL, cooling buffaloes twice daily at 07:00 and 16:00 h for 5 min each applying water with a handheld hosepipe; 3CS, three showering sessions of one hour each at 07:00, 11:00, and 15:00 h; 5CS, five showering sessions of one hour each at 07:00, 09:00, 11:00, 13:00, and 15:00 h. The showering was applied in cycles with 3 min water on and 3 min off using a flow rate of 2 L/min. ^2^ SEM = standard error of means.

**Table 4 animals-13-03315-t004:** Blood metabolites of Nili Ravi buffaloes were subjected to different cooling session treatments during a semi-arid summer.

Items	Cooling Sessions ^1^, n/d				
	CTL	3CS	5CS	SEM ^2^	*p* Value
Glucose, mg/dL	71.8 ^b^	83.9 ^a^	89.4 ^a^	4.0	0.006
Blood UREA Nitrogen, mg/dL	17.9 ^c^	14.2 ^b^	12.1 ^a^	0.45	0.001
Cortisol, µg/dL	5.5 ^b^	4.7 ^b^	3.8 ^a^	0.37	0.002

^a–c^ Values with different superscripts in a row are significantly different (*p* < 0.05). The values are presented as Least Square Means. ^1^ Cooling sessions treatment groups; CTL, cooling buffaloes twice daily at 07:00 and 16:00 h for 5 min each applying water with a handheld hosepipe; 3CS, three showering sessions of one hour each at 07:00, 11:00, and 15:00 h; 5CS, five showering sessions of one hour each at 07:00, 09:00, 11:00, 13:00, and 15:00 h. The showering was applied in cycles with 3 min water on and 3 min off using a flow rate of 2 L/min. ^2^ SEM = standard error of means.

## Data Availability

The data presented in this study are available on request from the corresponding author.

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
