# Peer review of "Heat Stress Mitigation: Impact of Increased Cooling Sessions on Milk Yield and Welfare of Dairy Buffaloes in a Semiarid Summer"

_animals, 2023, doi:10.3390/ani13213315_

Round 1

Reviewer 1 Report

The following suggestions are recommended:

Correct the notation of the hours throughout the manuscript, e.g. figure 1500 h, it should be placed 15:00 h.

Line 21: … to five times a day, on milk yield …

Line 23: … 2) 3CS, where buffaloes were …

Line 56: … sessions from two to three times a day resulted in …

Line 58: … The objective of the current study …

Line 62: … buffaloes in the challenging …

Line 71: … north sides

Line 72: … having open windows, and an …

Line 95: Check: Would it be Wecon instead of Wecam?

Line 105: … Lactoscan Standard, Milktronic lLtd. …

Line 114: … a digital …

Line 145:  … the mean value was 86,5 ± 4,2 SD.

Line 146:  Table 1. … Treatment Period (08:00 to 17:00 h) …

Line 159:  Table 2. LS Means ± SE = put the meaning below the table.

                   Items ... SEM = put the meaning below the table.

Line 161 - 165: Correct the notation of the hours, e.g. figure 1500 h, it should be placed 15:00 h.

Line 181 - 186: Correct the notation of the hours, e.g. figure 1500 h, it should be placed 15:00 h.

Line 184: … water with a handheld hosepipe …

Line 186: … The showering was applied in cycles with 3 min water on and 3 min off using

Line 187: … the cooling sessions timings. …

Line 191 - 192: … Recatal …

Line 196 - 198: Correct the notation of the hours, e.g. figure 1500 h, it should be placed 15:00 h.

Line 194: … cooling sessions strategies …

Line 196: … with a handheld hosepipe …

Line 199: … with 3 min water on and 3 min off using …

Line 215 - 218: Correct the notation of the hours, e.g. figure 1500 h, it should be placed 15:00 h.

Line 216: … with a handheld hosepipe …

Line 218: … with 3 min water on and 3 min off using …

Line 235: Place the meaning of LS Means ± SE; SEM in the same way as in the previous tables.

Line 237 – 239: Correct the notation of the hours, e.g. figure 1500 h, it should be placed 15:00 h.

Line 240: … with 3 min water on and 3 min off using …

Line 241: .3.1. Blood Metabolites

Line 254: Place the meaning of LS Means ± SE; SEM in the same way as in the previous tables.

Line 257 – 259: Correct the notation of the hours, e.g. figure 1500 h, it should be placed 15:00 h.

Line 256 and 258: Check the 8CS notation, is this wrong? Correct.

Line 292: … in line with the previous …

Line 307: … at different time points

Line 353: … that cows they preferentially …

The use of language is acceptable and understandable, the writing is correct and easy to read. Minor adjustments are proposed.

Author Response

Please find the response to the reviewer's comments in the attached file.

Reviewer 2 Report

Manuscript title: Stress Mitigation: Impact of Increased Cooling Sessions on Milk Yield and Welfare of Dairy Buffaloes in a Semiarid Summer.

Manuscript ID: animals-2632599

Journal: Animals (ISSN 2076-2615)

Section: Cattle

Spelcialty section: Water Buffalo Welfare, Strategies to Improve Health, Behavior, Productivity, and Food Quality and Safety

Article type:Original Research Article                                                                                                                   

Thank you for letting me review the manuscript titled: " Stress Mitigation: Impact of Increased Cooling Sessions on Milk Yield and Welfare of Dairy Buffaloes in a Semiarid Summer ". I consider that this manuscript could be published after some minor revisions that I addressed below.

General comments:

The Introduction and discussion sections need to be improved. I kindly request the authors to enhance both the introduction and the depth of analysis in the discussion.

Some references that are highly relevant and could help to write a more comprehensive examination of your article are listed below:

http://dx.doi.org/10.31893/jabb.21003

https://doi.org/10.3390/ani13172735

https://doi.org/10.3390/ani11082247

https://doi.org/10.3389/fvets.2022.963205

https://doi.org/10.3390/ani11061733

Title

Page 1: The title of the article accurately reflects the objective of the study.

Simple Summary

Page 1/Line 11-19: The authors meet the objective.

Abstract

Page 1/Line 20-33: This section adequately summarizes the aim, methodology, and results, concluding precisely. However, consider improving the methods and results part, indicating the P value and which blood metabolites showed improvement.

Introduction:

I would suggest improving the introduction section. Please, provide timely and relevant data with background on the study's topic. This will enrich this section and highlight the importance of the present findings.

For example:

Page 1/Line 39-40: I suggest being more specific and state what is the importance of these animals.

Page 1/Line 42: Consider mentioning the challenges that these animals face (they refer to changes in physiological variables, behavior, etc.). The authors could add values ​​of these variables or a physiological explanation of how this negatively affects the welfare and productivity of animals.

Page 2/Line 54-57: Please, rewrite this paragraph to improve its understanding to the reader.

Methods and materials:

In general, the Methods section is correctly structures. However, some details need to be added.

Page 3/Línea 98: Climate Measures. Indicate whether the environmental measurements were carried out every day throughout the six evaluation weeks.

Page 3/Línea 104-106: Specify the followed methods to obtain the milk sample for the composition analysis. 

Discussion

I kindly request the authors to enhance the depth of analysis in the discussion.

The authors present very interesting findings that could be discussed with a more in-depth physiological interpretation.

Reference:

While most of the references follow the Journal's style, please, revise the following examples and amend accordingly:

Page 10/Line 401: add a comma (,) after the publication year.

Page 11/Line 430: add the publication year.

Lastly, the following references are highly relevant and will aid in a more comprehensive examination of your article:

http://dx.doi.org/10.31893/jabb.21003

https://doi.org/10.3390/ani13172735

https://doi.org/10.3390/ani11082247

https://doi.org/10.3389/fvets.2022.963205

https://doi.org/10.3390/ani11061733

Author Response

(The authors gave the same response as above.)

Reviewer 3 Report

I appreciate the opportunity to review the present study focused on assessing the effect that cooling systems has on buffalo’s thermoregulation. As global warming is a current topic, studies towards aspects that might help animals to mitigate this issue are highly relevant. I made some comments hoping they can be useful for the authors.

Line 12. I recommend specifying how the authors evaluated “buffalo welfare” (e.g., through physiological parameters such as core temperature, respiratory rate, blood parameters, etc.).

Lines 16-17. Stating that blood metabolites improved is a little too general. I would recommend clearly mentioning which blood parameters (e.g., cortisol, glucose, etc.) and if they increased/decreased to be considered an improvement.

Line 21. Following my previous comment, consider specifying the parameters used to evaluate welfare.

Lines 42-43. Regarding the behavioral response of buffaloes, additional information about the time the animals stay inside the ponds or potholes, or the frequency of use would be appropriate. Also, while the behavioral response of animals to heat stress is relevant, other physiological alterations such as vasomotor changes and consequent alteration in the cardiorespiratory pattern could be mentioned as well, as mentioned by Marai and Haeeb in their revision focusing on water buffaloes (https://doi.org/10.1016/j.livsci.2009.08.001), or Mota-Rojas et al. revision regarding mammal thermoregulation and both behavior and physiological changes (https://doi.org/10.3390/ani11061733).

Line 46. Before mentioning the current mechanisms to prevent heat stress, I would recommend briefly mentioning the consequences of heat stress in those production systems where buffaloes are not provided by appropriate potholes or ponds to wallow.

Line 47. Please, define the concept of “effective showering” considering the thermoregulatory physiology of the species. A reference that might be helpful is 10.31893/jabb.23021.

Lines 52-54. Could the authors define how the use of sprinklers increases productive parameters on animals. For example, the factor that influences the most is the number of times these sprinklers are used, or its type, or the duration of the cooling method, etc. Please, try to include this.

Lines 58-61. I suggest defining the general aim of the study. As it is now, it seems that it has two aims.

Lines 68-69. I would suppose that the authors also included clinically healthy animals. If so, please, add if a physical exam was applied to the animals before including them in the present study.

Line 87. Please, include additional information about the sprinklers such as the brand, type, where were they placed, etc. This is important information if authors want to make these results reproducible for other researchers (and because the main topic of the present study is the implementation of sprinklers).

Line 145. Change “means ± SD” with “mean ± SD”.

Lines 273-274. A brief discussion of why heat stress can impair milk yield would be appropriate after these lines (e.g., the role of the hypothalamic-pituitary adrenal axis and stress hormones milk ejection on oxytocin release, which could also be related to the physiological response).

Line 287-288. As my previous comment, it is important to make this physiological association between stress and alterations in milk yield and milk composition.

Line 297. Consider adding a couple of references where it is reported that protein and lactose percentages can change according to the nutrition, genetics, or physiological responses of buffaloes.

Line 308-309. I recommend including that tachypnea is a compensatory mechanism of evaporative heat loss when animals are under heat stress. For example, the type of cooling systems that the authors compared might have a different cooling mechanism by aspersion (handheld hosepipe vs. sprinklers), causing the possible differences.

Line 362. Before the conclusions, the authors could add a section mentioning the limitations of the study and its application on dairy farms, according to the findings.

Author Response

(The authors gave the same response as above.)

Reviewer 4 Report

Heat Stress Mitigation: Impact of Increased Cooling Sessions 2 on Milk Yield and Welfare of Dairy Buffaloes in a Semiarid 3 Summer 4

Syed Israr Hussain1, Nisar Ahmed1, Saeed Ahmad2, Maqsood Akhter3 and Muhammad Qamer Shahid1,* 5

Lin 12: 18 cows for 3 tests is a very limited sample for this kind of experiment.

Line 69: what was the average day in parity? Days in milk and the daily milk yield of each group. What was the standard deviation for this average? Those question are answer on line 82-84. This should be in the same paragraph

Line 77: is the water was ab libitum during the day too?

Line 85: what time the day those 5 min applied in the CTL?

Line 88: what was the group average of each CS?

Line 92: why it was chosen for a full 1 hour and 6 min. cycle?

Line 93: what were the size of the sprinkler spreading? How far to the side did they mitigate?

Line 113: did the parameters were measured before cooling session started?

Line 114: what was the different between the people measuring the RR? Was it the same person taking the RR? not clear

Line 123: the blood sample was taken at 11:00? Why? How it corollate with the time of the CTL?

Line 167: The main different between the CBT was after 13:00 till 18:00 as presented in fig. 1 Did this time of day was assess for blood sampling?

244: not clear how cortisol was measured. There was a base line per cow before the experiment? Did each individual cow was compered to herself? Or it present the absolute parameters of cortisol?

Author Response

(The authors gave the same response as above.)
